# Immunotherapy for Treatment of Pleural Mesothelioma: Current and Emerging Therapeutic Strategies

**DOI:** 10.3390/ijms251910861

**Published:** 2024-10-09

**Authors:** Lauren Chiec, Debora S. Bruno

**Affiliations:** 1University Hospitals Seidman Cancer Center, Cleveland, OH 44106, USA; 2School of Medicine, Case Western Reserve University, Cleveland, OH 44106, USA

**Keywords:** mesothelioma, immunotherapy, immune checkpoint inhibitor

## Abstract

Pleural mesothelioma is a rare malignancy associated with asbestos exposure and very poor prognosis, with a 5-year overall survival of 12%. Outcomes may vary according to stage at time of diagnosis and histologic subtype. Most recently, clinical trials utilizing dual checkpoint inhibitor regimens and chemotherapy in combination with immune oncologic agents have demonstrated impactful changes in outcomes. In this article, we review studies that have led to the successful implementation of immunotherapy in clinical practice for the treatment of this disease and highlight ongoing clinical trials exploring the use of different immunotherapy strategies for the treatment of pleural mesothelioma. We also discuss the challenges of immunotherapy-based approaches in the context of mesothelioma and future strategies currently being investigated to overcome them.

## 1. Introduction

Pleural mesothelioma (PM) is a rare malignancy arising from the pleural surface and is associated with a poor prognosis. The primary risk factor for development of PM is inhalational asbestos exposure, often via occupational or environmental means, with a prolonged period of at least 15 and on average 30–40 years from the time of exposure to diagnosis [1,2]. In 2020, over 30,000 new cases of PM were identified globally, with the worldwide incidence of PM decreasing in recent years, most likely due to the increased regulations surrounding asbestos [3]. However, the incidence in several countries, specifically those with less restrictive regulation of asbestos use, is predicted to increase in the coming years [3,4].

In 2021, the World Health Organization (WHO) fifth edition classified PM into three histologic subtypes: epithelioid, sarcomatoid, and biphasic [5]. Epithelioid mesothelioma (EM) is the most common subtype, representing about 60% of all mesothelioma diagnoses [6]. The historical median survival of patients diagnosed with mesothelioma is associated with these subtypes, ranging from an average of 19 months for patients with epithelioid disease undergoing surgical resection to 4 months for those with sarcomatoid disease [7]. Despite advances in oncologic care, in 2018, the relative 5-year overall survival for all patients was around 14.6% [8].

Historically, platinum-based chemotherapy has been the standard of care for the systemic treatment of mesothelioma, with limited improvement in outcomes with the addition of novel agents and maintenance therapies. More recently, however, the use of immune-checkpoint inhibitors has been shown to significantly improve outcomes for many patients. This review will detail challenges and nuances in the diagnosis of mesothelioma, as well as provide an overview of outcomes of key mesothelioma clinical trials with a focus on the current and future role of immunotherapy and discussion of novel strategies for treatment.

## 2. Diagnosis

The diagnosis of mesothelioma can be challenging to determine based on pleural fluid analysis or small biopsies alone [9], and often, surgical biopsy via video-assisted thoracoscopic surgery (VATS) is pursued. Although imaging can provide important information regarding stage, surgical staging with thoracoscopy and mediastinoscopy to assess for nodal disease (or endobronchial ultrasound and biopsy) is the gold standard for patients who are eligible to undergo these procedures [10]. Surgical staging is particularly important for those patients being considered for surgical resection, as studies have shown that most patients with clinical staging alone are upstaged at the time of surgery based on pathologic evaluation [11].

In addition to staging information, histologic subtyping is crucial in understanding prognosis and determining optimal treatment. The epithelioid variant, representing about 60% of mesotheliomas, includes several architectural patterns such as tubulopapillary, trabecular, adenomatoid, micropapillary and solid epithelial patterns [7]. In addition, grading based on nuclear grade, mitotic count, and necrosis has been found to have prognostic significance [12,13], and the WHO 2021 guidelines recommend using these factors to classify epithelioid mesothelioma into low- or high-grade disease, recognizing the heterogeneity within the subtype in terms of prognosis [5]. *BAP1* is a tumor suppressor gene, and loss of BAP1 nuclear expression, tested by IHC, can be seen in about 70% of epithelioid mesotheliomas, aiding in diagnosis by differentiating between benign mesothelial hyperplasia and mesothelioma [14].

Sarcomatoid mesothelioma (SM) is diagnosed less often than epithelioid disease, and can be challenging to diagnose as *BAP1* loss is seen less frequently than in epithelioid disease. In more than 90% of cases, sarcomatoid mesothelioma is found to have a *CDKN2A* homozygous deletion or loss of methylthioadenosine phosphorylase gene (*MTAP*), and in most cases, co-deletion is seen. This is often identified by loss of expression of the MTAP protein product on immunohistochemistry as a surrogate marker for *CDKN2A* deletion, which can aid in making the diagnosis of SM [1,15]. Biphasic mesotheliomas are defined by disease involving at least 10% of both epithelioid and sarcomatoid components, with a true incidence rate being difficult to ascertain given the diagnostic challenges.

In general, tumor mutational burden in all subtypes of mesothelioma tends to be low [16,17], with <2 non-synonymous mutations per megabase seen in nearly all samples from one analysis of 74 samples of primary mesothelioma from The Cancer Genome Atlas [16]. In addition to mutations in *BAP1*, *CDKN2A* and *MTAP*, other somatic mutations frequently seen in mesothelioma include mutations in the tumor suppressors *NF2*, *TP53*, *LATS2* and *SETD2* [16].

To date, no blood-based biomarker has been reliable for use as a screening or diagnostic tool for mesothelioma. Mesothelin, a glycoprotein with a role in cell adhesion and which is overexpressed in mesothelioma, can be detected in the blood. Although the role of mesothelin for screening or diagnosis of mesothelioma remains limited, it has been shown to correlate with disease status and may eventually play a role in monitoring response to treatment [1].

## 3. Systemic Treatment of Advanced Disease—Chemotherapy, Vascular-Endothelial Growth Factor and Other Non-Immunotherapy Strategies

Historically, the standard of care for patients with unresectable mesothelioma has been cytotoxic chemotherapy, with the combination of cisplatin and pemetrexed, based on the results of the EMPHACIS trial. This study evaluated 456 patients treated with cisplatin and either pemetrexed or placebo, once every three weeks [18]. The addition of pemetrexed to cisplatin improved the median overall survival (12.1 versus 9.3 months, HR 0.77, *p* = 0.020) as well as the objective response rate (41 versus 17%, *p* < 0.0001) [18]. Subsequently, a separate study evaluated the role of maintenance pemetrexed compared with observation alone after the completion of four to six cycles of platinum chemotherapy and pemetrexed. A total of 49 patients without progressive disease after this initial therapy were enrolled. Median progression-free survival was similar between groups (3.4 versus 3 months, HR 0.99, 95% CI 0.51–1.90, *p* = 0.9733) with a difference in median overall survival (16.3 versus 11.8 months, HR 0.86, 95% CI 0.44–1.71, *p* = 0.6737) that was not statistically significant [19]. Although the trial was likely not able to detect a benefit, some experts do recommend maintenance pemetrexed.

Other maintenance therapy strategies have also not been shown to provide an overall survival benefit. Several studies evaluated the role of maintenance gemcitabine for patients without disease progression on first-line chemotherapy. A phase 2 NVALT19 study enrolled 130 patients who were randomized after completion of at least four cycles of first-line platinum and pemetrexed chemotherapy and had no evidence of disease progression, to receive either gemcitabine plus supportive care or supportive care alone. In the group who received gemcitabine, PFS was significantly longer than the group receiving supportive care alone (median PFS 6.2 months vs. 3.2 months, HR 0.48, 95% CI 0.33–0.71, *p* = 0.0002). The phase 2 GEMO study enrolled 64 patients without disease progression after 4–6 cycles of either platinum–gemcitabine or platinum–pemetrexed chemotherapy. Patients were randomized to receive either gemcitabine or best supportive care until disease progression or unacceptable toxicity. Although PFS was significantly higher in the patients receiving gemcitabine maintenance therapy (median PFS 6.2 months vs. 2.8 months, *p* < 0.001), there was no significant difference in OS between the groups (23.3 months vs. 13.4 months, *p* = 0.155).

Efforts to improve on the standard chemotherapy backbone have been underway for decades, including the addition of vascular endothelial growth factor (VEGF)-targeted therapy such as bevacizumab. In the phase III randomized MAPS study of 448 patients, the addition of bevacizumab to cisplatin and pemetrexed (with bevacizumab maintenance after completion of six cycles of chemotherapy) showed an improvement in median OS (18.8 versus 16.1 months; HR 0.77, 95% CI 0.62–0.95) compared with cisplatin and pemetrexed alone [20]. This has led to many experts utilizing bevacizumab with platinum-based chemotherapy. However, conflicting results were seen in a phase II trial of bevacizumab maintenance after carboplatin and pemetrexed [21], which failed to meet its primary endpoint of at least a 50% improvement in median PFS compared to the historical standard of 6 to 9 months.

Other studies have also explored the use of antiangiogenic agents in combination with standard chemotherapy. The phase II SWOG S0905 study evaluated the use of cediranib, an inhibitor of both VEGF-receptor and platelet-derived growth factor receptor (PDGFR) in 92 patients randomized to receive either cediranib or placebo along with standard platinum-pemetrexed chemotherapy in the first-line setting [22]. Cediranib was shown to improve PFS (7.2 months vs. 5.6 months, HR 0.71, 80% CI 0.54–0.95, *p* = 0.062) and increased modified RECIST v1.1 response (50% vs. 20%, *p* = 0.006); however, no significant difference in overall survival was seen [22]. In addition, there was a significantly higher rate of grade 3 and 4 diarrhea, dehydration, hypertension, and weight loss in the cediranib arm compared to the control arm, limiting further development of cediranib in mesothelioma.

The phase II LUME-Meso trial evaluated 87 patients randomized to treatment with nintedanib, a small molecule tyrosine kinase inhibitor targeting VEGF receptors, PDGFR, and fibroblast growth factor receptors, in combination with cisplatin and pemetrexed or to chemotherapy alone. The study showed an improvement in PFS (9.7 months vs. 5.7 months, HR 0.54, 95% CI 0.33–0.87, *p* = 0.010) and OS (20.6 months vs. 15.2 months, HR 0.49, 95% CI 0.30–0.82, *p* = 0.006) in the group treated with nintedanib [23]. However, in the subsequent phase 3 trial of 458 patients randomly assigned to the two groups, no significant difference in PFS was seen, with a median PFS of 6.8 months in the nintedanib group and 7 months in the placebo group (HR 1.01, 95% CI 0.79–1.30, *p* = 0.91) [24].

Mesothelioma has also been identified as a cancer with frequent loss of arginosuccinate synthetase 1 (*ASS1*), which is a tumor suppressor gene and urea cycle enzyme. The loss leads to a reliance by the cancer cells on arginine for survival, prompting investigation into arginine deprivation strategies [25]. An initial phase II trial, ADAM (Arginine Deiminase and Mesothelioma) randomized patients with advanced mesothelioma to receive either pegargiminase, which degrades arginine, with best supportive care, or to best supportive care alone. It demonstrated a 1.2-month PFS benefit for the study arm [26]. Subsequently, the phase I TRAP study combined pegargiminase with cisplatin and pemetrexed for patients with ASS1-deficient thoracic cancer, and was found to be safe and had a greater than 90% disease control rate (DCR) including patients with pleural mesothelioma, with a median OS of 10–14 months [27,28]. Notably, this included patients with nonepithelioid subtype, where chemotherapy alone has historically provided minimal benefit.

The phase III ATOMIC-Meso randomized trial enrolled 249 chemotherapy-naïve patients with nonepithelioid pleural mesothelioma, and compared pegargiminase in combination with platinum and pemetrexed chemotherapy with chemotherapy alone. The median OS for patients in the pegargiminase arm was 9.3 months, compared with 7.7 months in the control arm (HR 0.71, 95% CI 0.55–0.93, *p* = 0.02) [25]. This may represent a new treatment strategy for a historically difficult to treat population.

## 4. Immunotherapy—Relapsed/Refractory Disease

The development of immunotherapy, specifically immune-checkpoint inhibitors, and the success in other tumor types led to interest in developing these therapies in mesothelioma. Retrospective studies have shown that high tumor expression of programmed cell death-ligand 1 (PD-L1), known to inhibit T-cell function when bound to the programmed cell death-1 (PD-1) protein, is associated with poor prognosis in patients with mesothelioma [29,30]. In addition, moderate- to high-intensity PD-L1 staining (at least 50% of tumor cells) has been correlated with non-epithelioid histology [31].

### 4.1. Tremilimumab

Despite the initial enthusiasm for this strategy, preliminary studies evaluating the role of immune-checkpoint inhibition, utilizing single-agent strategies after progression on first-line platinum-based chemotherapy yielded disappointing results [32,33]. The phase II MESOT-TREM-2008 study was a single-arm study of the cytotoxic T-lymphocyte associated protein 4 (CTLA-4) targeting agent tremilimumab, dosed at 15 mg/kg once every 90 days, for patients with progressive disease after standard first-line platinum–pemetrexed therapy. Only 2 out of 29 patients showed an objective response; however, the response seen in those two patients lasted over 6 months [34]. Subsequently the MESOT-TREM-2012 study, which utilized an “intensified” dosing strategy of 10 mg/kg every 4 weeks, initially also failed to show a consistent benefit in terms of overall response rate (ORR) or DCR, but interestingly, did show a median OS of 15.8 months in the seven patients with biphasic or sarcomatoid histologies, a significant improvement compared to historical controls [35]. These studies ultimately led to the phase IIb DETERMINE trial, which enrolled 571 patients with both unresectable pleural and peritoneal mesothelioma and progression after one or two previous systemic therapies for advanced disease, and randomized them in a 2:1 fashion to receive either tremelimumab (10 mg/kg every 4 weeks to start) or placebo. However, no significant difference in median OS was seen (7.7 vs. 7.3 months, HR 0.92, 95% CI 0.76–1.12, *p* = 0.41) [32].

### 4.2. Pembrolizumab

In addition to studies evaluating the role of CTLA-4 targeting immune-checkpoint inhibition, immune-checkpoint inhibitors targeting PD-1 and PD-L1 have been explored in the refractory setting. In the phase Ib KEYNOTE-028 study, patients with disease progression after standard systemic therapy and with PD-L1 positive disease were enrolled as part of a basket trial to receive the PD-1 inhibitor pembrolizumab. In this study, pembrolizumab was associated with a 20% ORR, with 52% of patients having stable disease (SD) and a median duration of response (DOR) of 12 months. It was noted that 4 of the 25 patients had a >70% decrease in tumor burden, a finding not reported previously with the use of tremelimumab. In addition, exploratory analysis found that PD-L1 expression and/or tumor mutational burden was associated with a higher likelihood of response to treatment [36].

Subsequently, additional phase 2 studies of previously treated patients revealed similar trends. The KEYNOTE-139 study enrolled 65 patients treated with pembrolizumab after 1–2 prior lines of therapy and found a DCR of 63% at 9 weeks. Higher expression of PD-L1 correlated with higher response rate and PFS, but not OS [37,38]. KEYNOTE-158, a large basket trial of pembrolizumab after progression on standard systemic therapy, included 118 patients with mesothelioma. ORR was 8%; however two-thirds of responders showed an ongoing response at 12 months with a median DOR Of 14.3 months [39].

PROMISE-meso was a randomized phase III study of pembrolizumab compared to standard chemotherapy (gemcitabine or vinorelbine) for patients with progression after platinum-based chemotherapy. Patients (*n* = 73) were randomized in a 1:1 fashion, and crossover was allowed. Ultimately, results from the interim analysis were disappointing, with a median PFS of 2.5 months for those patients receiving pembrolizumab and 3.4 months for those receiving chemotherapy (HR 1.06, 95% CI 0.73–1.53, *p* = 0.76) [33]. Despite adjustment for crossover (63% in the chemotherapy arm), no significant survival benefit was seen with pembrolizumab. PD-L1 status was not associated with efficacy; however, there was a significant proportion of patients with missing data (>25%).

### 4.3. Nivolumab

The anti-PD-1 inhibitor nivolumab has also been evaluated for patients with disease progression after initial treatment in several phase II trials. The NivoMes trial met its primary endpoint and showed an ORR of 24% and DCR of 29% at 6 months [40]. The MERIT trial (*n* = 34) showed an ORR of 29.4% with a DCR of 67.6% at 6 months [41]. At median follow-up of 16.8 months, the median PFS was 6.1 months and median OS 17.3 months. Both PFS and OS benefit correlated with PD-L1 positivity, and 3-year follow-up revealed 8 of the 34 patients were alive at 3 years of follow-up [42]. Based on this study, nivolumab was approved in Japan for patients with progressive disease after initial platinum-based chemotherapy.

Subsequently, the phase 3 CONFIRM trial (*n* = 332) randomized patients in a 2:1 fashion to receive treatment with nivolumab or placebo after platinum-based chemotherapy. The nivolumab arm showed a significant improvement in overall survival, with median OS at 17 months of 9.2 months compared with 6.6 months (HR 0.72, 95% CI 0.55–0.94, *p* = 0.02) [43,44].

### 4.4. Avelumab

The phase 1b JAVELIN basket study evaluated the role of avelumab, an anti-PD-L1 targeting agent, for patients after previous treatment with platinum and pemetrexed-based chemotherapy and included 53 patients with mesothelioma, after up to three lines of prior therapy [45]. It demonstrated a 9.4% ORR (18.8% in those with tumor PD-L1 > 5%), with a DCR of 58.5% and a median DOR of 15.2 months. The median OS was 10.9 months.

### 4.5. Dual Immune-Checkpoint Inhibition

With initial signals of activity, including some durable responses with immune-checkpoint inhibitor monotherapy, investigators next sought to evaluate the role of dual immune-checkpoint inhibition, targeting both the CTLA-4 and PD-1/PD-L1 checkpoints. MAPS2 was a phase II study that evaluated the role of nivolumab or the combination of ipilimumab and nivolumab for patients previously treated with one to two lines of systemic therapy, including platinum-based chemotherapy [46]. Although this study (*n* = 125) was non-comparative, it reported an ORR of 19% and median OS of 11.9 months (95% CI 6.7–17.4) for patients in the nivolumab arm and ORR of 26–28% with median OS of 15.9 months (95% CI 10.7–22.2) for patients in the combination arm. It met the primary endpoint of DCR, with 44% and 50% of the patients in the nivolumab and nivolumab plus ipilimumab arms, respectively, achieving either stable disease or an objective response over 12 weeks. Notably, ORR and DCR increased significantly in patients with high PD-L1 (tumor > 25%). Three patients in the nivolumab arm and 10 patients in the combination arm had tumor burden shrinkage of greater than 60%. Two-year OS was 25.4% in the nivolumab arm (95% CI 15.5–36.6) and 31.7% in the combination arm (95% CI 20.5–43.4).

The smaller but similar, non-randomized INITIATE trial was a phase II study of 35 patients after receipt of at least one prior line of chemotherapy (consisting of platinum-based chemotherapy), who were treated with the combination of ipilimumab and nivolumab. The ORR at 12 weeks was 29% with a DCR of 68% [47], in similar fashion to what MAPS2 had reported.

NIBIT-MESO-1 was a phase II study including patients with progressive disease after first-line platinum-based chemotherapy (*n* = 28) as well as those refusing first-line chemotherapy (*n* = 12). Patients received a combination of tremelimumab and durvalumab, with a reported median OS of 16.6 months and a 65% DCR. Interestingly, however, PD-L1 expression did not provide predictive or prognostic information [48].

The abundance of studies evaluating the role of immunotherapy for patients with recurrent or progressive mesothelioma clearly show a signal for response for a subset of patients, with some responders showing durable long-term responses. Given these findings, research has subsequently focused on attempting to utilize immunotherapy earlier in the disease process, identifying predictors of response to immunotherapy and novel combinations including strategies to enhance in the immune response. Table 1 summarizes key studies of immunotherapy for patients with progressive disease after initial treatment.

## 5. Immunotherapy—First-Line

The phase 3 CheckMate 743 study, which randomized 605 patients with unresectable mesothelioma to first-line platinum-based chemotherapy or the combination of nivolumab and ipilimumab, was the first to show a significant improvement in median OS with the use of first-line immunotherapy, with a median OS of 14.1 months with standard chemotherapy and 18.1 months with dual immunotherapy (HR 0.74, 96.6% CI 0.60–0.91, *p* = 0.002) [49]. This benefit was seen despite 20% of patients on the chemotherapy arm going on to receive immunotherapy in a later line. Notably, while OS with dual immunotherapy was improved regardless of histologic subtype, the most marked improvement was seen in those with non-epithelioid histology (median OS in epithelioid patients 18.7 months in immunotherapy arm vs. 16.5 months in chemotherapy arm; non-epithelioid patients 18.1 months vs. 8.5 months). This is consistent with the lack of response to standard chemotherapy for patients with non-epithelioid histology reported in prior studies.

Other notable findings from this study include the observation that although there seemed to be similar ORR between chemotherapy and immunotherapy (43 vs. 40%), more patients treated with immunotherapy had durable responses (32% with ongoing response at 2 years, compared to 8% in the chemotherapy group), consistent with early trials. In an exploratory analysis, a four-gene inflammatory signature including PD-L1, LAG-3, CD8a and STAT1 with a “high” compared to “low” score was associated with improved OS for patients treated with immunotherapy (OS 21.8 months vs. 16.8 months, HR 0.57, 95% CI 0.40–0.82), but not patients treated with chemotherapy (OS 11.6 months vs. 15.2 months, HR 1.14, 95% CI 0.82–1.59) [50].

Although patients treated with immunotherapy had up to twice as many treatment-related adverse events as those treated with chemotherapy, the survival benefit at 1 and 2 years remained consistently higher in the immunotherapy group. Patient-reported outcome measures also favored the immunotherapy arm [51]. Based on this study, the combination of ipilimumab and nivolumab is recommended for the first-line treatment of pleural mesothelioma and is certainly preferable over chemotherapy for those with non-epithelioid histology if there are no contraindications [50].

Several additional studies have been completed to assess the efficacy of first-line immunotherapy in combination with platinum-based chemotherapy for patients with mesothelioma. The phase II DREAM study reported on outcomes for 54 patients treated in Australia with first-line cisplatin, pemetrexed, and durvalumab. The ORR was 48% and DCR 85%, with six patients reported to have tumor shrinkage of over 80% [52]. With a median of 28 months of follow-up, median OS was reported at 18.4 months (95% CI 13.1–24.8) with a 24-month OS of 37% (95% CI 26–52%). Similarly, the phase II PrE0505 study of 55 patients treated with first-line platinum and pemetrexed chemotherapy with durvalumab reported an ORR of 56.4% and median OS of 20.4 months at median follow-up of 24 months (HR 0.34 compared with historical controls, *p* = 0.0014) [53]. The phase III DREAM3R trial is ongoing and evaluating the role of durvalumab combined with platinum and pemetrexed chemotherapy, compared with chemotherapy alone [54]. The trial plans to recruit 480 patients through 2024 with OS as the primary endpoint.

The JME-001 trial, a phase II trial of first-line cisplatin, pemetrexed, and nivolumab for patients (*n* = 18) with unresectable mesothelioma reported an ORR of 77.8% (the study’s primary outcome) and median OS of 20.8 months [55]. The phase II-III IND227-IFCT1901-KEYNOTE-483 trial evaluated the role of first-line pembrolizumab combined with platinum and pemetrexed chemotherapy, compared with chemotherapy alone. A total of 440 patients were randomized with a median OS of 17.3 months in the pembrolizumab arm compared with 16.1 months in the chemotherapy alone arm (HR 0.79, 95% CI 0.64–0.98, *p* = 0.0324) [56]. Median PFS was 7.13 compared to 7.16 months, respectively (HR 0.80, 95% CI 0.65–0.99, *p* = 0.0372). An exploratory analysis demonstrated the benefit of adding immunotherapy to chemotherapy to be again mostly restricted to the non-epithelioid histology (OS 12.3 months in the pembrolizumab arm versus 8.21 months for chemotherapy alone arm, HR 0.57, 95% CI 0.36–0.89). In the epithelioid histology strata, OS was 18.2 months for the pembrolizumab arm versus 19.8 months for the chemotherapy only arm (HR 0.89, 95% CI 0.7–1.13).

Given previous findings from the MAPS trial using the VEGF targeted agent, bevacizumab, the phase III BEAT-meso study was designed. It randomized 400 patients in a 1:1 manner to receive either standard platinum-based chemotherapy with bevacizumab, or the same combination with the addition of atezolizumab. At median follow-up of 35 months, the median PFS was significantly longer in the atezolizumab arm (9.2 months vs. 7.6 months, HR 0.72, 95% CI 0.5–0.89, *p* = 0.0021). However, the median OS was 20.5 months in the atezolizumab arm, compared with 18.1 months in the control arm (HR 0.84, 95% CI 0.66–1.06, *p* = 0.14), a finding that was not statistically significant [57].

Combining established checkpoint inhibitors with new molecules such as antibody–drug conjugates (ADC) bearing a cytotoxic payload is also a strategy under investigation. Anetumab ravtansine is a mesothelin-targeting ADC linked to a microtubule inhibitor maytansinoid DM4. Its activity and safety in combination with pembrolizumab was assessed in a Phase I/II trial of patients with progression of disease after prior cytotoxic chemotherapy. No safety concerns were identified, and the study demonstrated a trend towards superior PFS that did not reach statistical significance due to lack of power (12.2 months or ADC combined with pembrolizumab versus 3.9 months for single-agent pembrolizumab; HR 0.55, *p* = 0.20). Interestingly, high soluble mesothelin levels were associated with lack of benefit for the combination [58].

There is clearly a benefit with the use of immunotherapy in the first-line setting, with consistent improvements in survival seen compared with standard chemotherapeutic approaches. However, as with initial studies evaluating use for patients with refractory disease, further research into biomarkers of response to identify which patients benefit most, and strategies to enhance the immune response to treatment are needed. In addition, optimal combination strategies of immunotherapy with chemotherapy remain unclear. Table 2 summarizes key studies of immunotherapy in the first-line setting.

## 6. Novel Immunotherapeutic Strategies and Cellular Therapies

In addition to PD-1 and CTLA-4, other strategies for leveraging the immune system are being explored and represent promising and novel approaches. VISTA (V-domain Ig-containing suppressor of T-cell activation) is a gene structurally like PD-L1 and when overexpressed, for example on T lymphocytes, leads to the suppression of T cell activity [59]. Interestingly, VISTA acts as a ligand on antigen-presenting cells as well as a receptor on T-cells. VSIG-3 is the ligand for VISTA. Compared with other tumor types, the Tumor Cancer Genome Atlas demonstrated increased mRNA expression of VISTA in pleural mesothelioma, making it an intriguing potential target for treatment. However, in a phase I study of 12 patients treated with CA-170, a small molecule inhibitor targeting VISTA as well as PD-1/PD-L1 pathways, no objective responses were seen, although prior immunotherapy was allowed [60].

LAG-3 is a T-cell inhibitory receptor that suppresses both T-cell activation and cytokine secretion, and ultimately plays a role in T-cell exhaustion [61]. Although no expression of LAG-3 is typically seen on mesothelial cells, high levels have been reported in pleural effusions of patients with mesothelioma [62,63]. Targeting LAG-3 is an active area of investigation in the treatment of mesothelioma; the bispecific molecule tebotelimab, which targets PD-1 and LAG-3, was studied in a phase 1 trial of patients with both solid tumors and hematologic cancers, including mesothelioma, and was found to be safe and warrants further investigation [61]. A phase 2 study evaluated the combination of ieramilimab, a humanized anti-LAG-3 antibody with spartalizumab (an anti-PD-1 antibody) for patients with select advanced solid malignancies, including mesothelioma [64]. A total of 57 patients with mesothelioma were enrolled, and of those naïve to treatment with anti-PD-L1 or anti-PD-1 therapy, the ORR was 17.1% (90% CI 8.3–29.7%), with a median PFS of 5.5 months (Table 1).

The role of epigenetic changes in response to immunotherapy and the development of resistance has also been explored. Studies have shown that epigenetic changes can alter gene expression related to antigen presentation as well as directly affect immune cell function, ultimately leading to inhibition of the immune response [65,66,67]. Initial preclinical and mouse studies revealed that drugs such as DNA hypomethylating agents (DHAs) and histone deacetylase inhibitors (HDACi) can lead to epigenetic remodeling of cancer cells and improve immune recognition [67,68,69,70,71,72,73,74]. In a phase 1 study of patients with mesothelioma treated with the HDACi vorinostat (suberoylanilide hydroxamic acid; SAHA), a partial response was seen in 2/13 patients. However, in the phase III VANTAGE-014 study of 650 patients treated with vorinostat after prior chemotherapy, no OS benefit from voristat was seen compared with placebo (HR 0.98, 95% CI 0.83–1.17, *p* = 0.86) [75].

### Cellular Therapies

Cellular therapies such as chimeric antigen receptor T-cell therapy (CAR-T) and dendritic cell therapies are being explored given the success of cellular therapy to date, specifically for hematologic malignancies. CAR-T therapy involves the creation of genetically modified T-cells that include a chimeric antigen receptor (or CAR); this receptor activates the T-cells against a tumor-specific antigen leading to an anti-tumor immune response. Preclinical studies have investigated several targets for use in mesothelioma including mesothelin (MSLN), FAP, cMET, and pan-ERbB [67]. Three patients were included in a phase I trial of MSLN-targeted CARS; however, all three patients developed an antibody response to the murine component of the CAR [76]. In a separate phase I study, investigators used a second-generation CD28-costimulated MSLN CAR with the icaspase-9 safety gene administered intrapleurally to 25 mesothelioma patients after at least one prior line of treatment [77]. A total of 18 of the patients received immunotherapy targeting PD-1 off study after intrapleural CAR-T. Of those treated with pembrolizumab, 2 patients demonstrated a complete metabolic response on PET, and 8 patients had stable disease for over 6 months. OS after CAR T-cell infusion was 23.9 months. A phase I trial of an anti-CD28-costimulated CAR targeting fibroblast activating protein (FAP), a cell-surface antigen highly expressed in epithelial cancers, enrolled three patients with mesothelioma and at least two prior lines of treatment, and delivered therapy intrapleurally. It was found to be safe, with two of the three patients alive at 18 months after treatment [78]. Further studies are ongoing, including the phase I/II EVEREST-2 study evaluating a logic-gated, MSLN-targeted Tmod CAR T-cell therapy that combines both an activating and blocking receptor to attempt to address on-target, off-tumor toxicity [79]. There is also an ongoing phase I/II study of the MSLN-targeted CAR-T therapy, gavocabtagene autoleucel (gavo-cel), which fuses the anti-mesothelin antibody to a glycine/serine spacer and human CD3ε subunit, which is then cloned into a lentiviral backbone. Preliminary results of 32 patients treated with gavo-cel showed an ORR of 20% (13% confirmed), DCR of 77%, and 6-month OS of 70% [80].

Dendritic cell therapy represents a cell-based vaccination approach to stimulate an anti-tumor response. It involves harvesting dendritic cells (or precursors), which are highly efficient at T-cell stimulation, from a patient’s blood or bone marrow, to be differentiated ex vivo and exposed to cancer antigens in vitro [67,81]. These cells are then injected back into the patient to stimulate a tumoral immune response through a variety of pathways.

In a phase I trial of 10 patients with mesothelioma, DC vaccination was found to be safe and resulted in a median OS of 19 months in patients with newly diagnosed mesothelioma when used as consolidative therapy after chemotherapy [65]. Similar results were seen in the phase I MesocancerVA trial including nine patients [82]. Subsequently, investigators completed a study of 10 additional patients with mesothelioma who were treated with consolidative DC vaccination in combination with low-dose cyclophosphamide to attempt to reduce the number of circulating T regulatory cells and improve immune response [83]. Here, 7 out of 10 patients treated survived greater than 24 months with several patients having promising long-term responses.

Subsequently, the phase II/III DENIM trial randomized 176 patients to receive DC therapy with up to five infusions of allogeneic tumor cell lysate with best supportive care, or to best supportive care alone as maintenance therapy after chemotherapy for patients with unresectable mesothelioma. However, at a median follow-up of 15.1 months, the median survival for the study arm was 16.8 months compared with 18.3 months in the best supportive care arm (HR 1.10, 95% CI 0.77–1.57, *p* = 0.62) [84]. Both groups went on to receive immune checkpoint inhibitor therapy at similar rates, and the authors concluded that future studies investigating combinations of DC vaccination therapy and immune checkpoint inhibitor therapy were warranted.

To this end, the phase IB MESOVAX trial is investigating the combination of DC vaccine therapy and pembrolizumab for patients with advanced pleural and peritoneal mesothelioma and previously treated disease. Interim results of the first six patients treated with the protocol, with a median follow-up of 20.1 months, demonstrated no grade 3–4 treatment-related adverse events (TRAEs) [85]. At the first tumor response analysis after 2.1 months, two patients had stable disease, one had a partial response, and three had progressive disease.

## 7. Other Strategies for Advanced Disease

Several other strategies have been developed in attempts to improve outcomes for patients with advanced pleural mesothelioma. Tumor treating fields (TTFs) involving the use of alternating electrical fields within the torso to disrupt cancer cell division and lead to apoptosis were evaluated in the phase II STELLAR trial, used in combination with platinum chemotherapy and pemetrexed [86]. A total of 80 patients were treated with a median OS of 18.2 months. Based on this study, the NovoTTF system was approved in 2019, although its routine use has been limited by the non-randomized nature of the study.

There have been attempts to assess the role of intrapleural therapies in the treatment of mesothelioma. One study utilizing intrapleural HSV1716 virus therapy in 13 patients was found to be safe [87]; however, intrapleural bacterial immunotherapy (TILT) was not found to be feasible [88]. There is an ongoing non-randomized phase I/II trial, MITOPE, assessing the role of intra-pleural RSO-021, a mitochondrial peroxide scavenging enzyme (peroxiredoxin 3 inhibitor) that causes oxidative stress and cell death. In an initial subset of 15 patients treated, it was found to be safe with a DCR of 70% in 10 evaluable patients, and 2 patients having disease control for over 30 weeks [89]. The phase II component of the study is ongoing.

Among other trials that have attempted to harness intrapleural therapy, ONCOS-102, an oncolytic adenovirus which expresses granulocyte-macrophage colony-stimulating factor, was combined with platinum-pemetrexed chemotherapy. ONCOS-102 is hypothesized to be a potent stimulator of the anti-tumor immune response. In 31 patients (25 randomized) to receive ONCOS-102 with chemotherapy or chemotherapy alone, the study arm was found to be well-tolerated and led to increased T-cell infiltration [90]. Although there was a trend towards improvement in OS for chemotherapy-naïve patients (*n* = 17; median OS 20.3 versus 13.5 months, *p* = 0.088), this was exploratory and not statistically significant. Future study combining ONCOS-102 with immune-checkpoint inhibitor therapy may be warranted.

The phase II NIPU study randomized 118 patients with disease progression after first-line therapy to receive both nivolumab and ipilimumab alone, or combined with UV1, a vaccine directed against telomerase [91]. The rationale for this trial is to utilize a vaccine targeted towards a tumor-related antigen to increase lymphocyte infiltration and, ultimately, immune response. The primary endpoint of PFS was not met, with a median PFS of 4.2 months for those in the study arm, compared with 4.7 months in the control arm (HR 1.01, 80% CI 0.75–1.36, *p* = 0.979). However, there was a trend towards improved ORR (31% compared to 16%, OR 2.44, 80% CI 1.35–4.49, *p* = 0.056) and overall survival (median 15.4 versus 11.1 months, HR 0.73, 80% CI 0.53–1.0, *p* = 0.197).

## 8. Early-Stage Disease

Historically, surgery has been offered to select patients, as part of a multimodal approach, with the optimal surgical approach being based on large institutional reports and prospective registries [92,93,94]. These include extrapleural pneumonectomy (EPP) or extended pleurectomy/decortication (P/D) with the goal of obtaining a macroscopic complete resection. Although definitions of P/D have been inconsistent in the past, extended P/D in the current era is ultimately that which involves parietal and visceral pleurectomy to remove all gross tumor, with resection of the diaphragm and/or pericardium as required [95]. Clinical trials have attempted to identify the optimal surgical approach and confirm a survival benefit with the use of surgery, without clear evidence for or against but with limitations which continue to leave questions surrounding the specific role for surgery [96,97].

Although there is a lack of randomized data to support the use of either adjuvant or neoadjuvant chemotherapy as part of a multimodality approach to treatment, most guidelines and experts recommend the use of up to four cycles of cisplatin and pemetrexed either before or after surgery, with real-world reports supporting the promise of this approach [98]. Given the improvement in outcomes for patients with unresectable disease with the use of immunotherapy, studies are ongoing to investigate the role of immunotherapy for patients with potentially resectable disease [99].

One such phase II window of opportunity study randomized patients with resectable mesothelioma to receive either neoadjuvant durvalumab or neoadjuvant durvalumab plus tremelimumab, followed by surgery. The primary outcome was alteration of the intratumoral CD8/regulatory T cell ratio after treatment; both groups had increase in CD8 T-cell infiltration into tumors, but no alteration of CD8/Treg ratios [100]. A total of 17/20 patients receiving immunotherapy went on to planned surgery. At 34.1 months of follow-up, the median OS was not reached for patients receiving dual checkpoint inhibitor therapy, compared with 14 months for those receiving durvalumab alone (*p* = 0.040).

There are new and ongoing trials (mostly actively recruiting patients) investigating the use of checkpoint inhibitors in the neoadjuvant setting for pleural mesothelioma (Table 3). This strategy is explored also in patients with sarcomatoid histology, which has been historically excluded from surgical trials due to its poor associated outcome and survival as well as lack of benefit from previously available systemic therapies. Additionally, the ongoing phase III AtezoMeso study will assess the role of adjuvant atezolizumab [101].

## 9. Discussion and Future Direction

Despite the significant improvements in outcomes with the use of immunotherapy, the overall response rate remains <50% at large, and biomarkers of response are needed to help identify which patients are most likely to benefit. As reported in the studies discussed previously, PD-L1, used as a biomarker of response in many other solid-tumor malignancies, has had mixed outcomes as a predictive or prognostic biomarker in mesothelioma. Similarly, tumor mutational burden has not shown to be a clear biomarker. Other immune-checkpoint molecules, such as T-cell immunoglobulin and mucin domains containing protein 3 (TIM-3) and VISTA, are being explored but likely represent only small contributors to a complex milieu involved in the immunologic response [102].

Given that not all patients with mesothelioma will have a response to currently approved immunotherapies, there is a need for novel treatment strategies for those patients, several of which are under active investigation. One such strategy involves the use of a combination of an autologous Wilms tumor 1 (WT1) dendritic cell vaccine and the anti-PD-L1 antibody atezolizumab, with standard platinum-pemetrexed chemotherapy, based on preliminary studies. Early studies have shown that the transcription factor WT1 is overexpressed in mesothelioma and showed early promise based on the ability of the vaccine to induce immune responses in patients [103]. Given studies to-date showing the potential synergistic effects of chemotherapy and immunotherapy, the ongoing phase I/II study will assess whether the addition of both immunotherapy and WT1 dendritic cell vaccination is feasible and safe in the first-line setting for patients with epithelioid mesothelioma [104].

Another novel strategy currently being investigated in a phase I/II clinical trial is assessing the safety of MRTX1719, which binds to PRMT5/MTA complex [105]. For patients with homozygous deletion of *MTAP* gene, there is an increase in MTA concentration and therefore an increased reliance on PRMT5 [106]. It is hypothesized that this cooperative inhibitor may selectively target cancer cells with *MTAP* deletion and spare normal tissue.

Despite initial successes with the use of dendritic cell and CAR-T cell therapies, challenges remain including the time and resource-consuming nature of these therapies and the need for generation of sufficient amounts at a specified quality to receive treatment. In addition, further investigation into optimal conditioning chemotherapy regimens is needed. The implications of the tumor microenvironment, ability of the CAR-T cells to traffic into tumor tissue, heterogeneity of the target antigens and overall response durability also remain to be seen [107].

## 10. Conclusions

The use of immunotherapy in patients with advanced pleural mesothelioma is currently a safe and established approach, and dual checkpoint inhibition should be offered to patients in the first-line setting, especially to those with non-epithelioid histology. Immunotherapy is also being actively investigated in the neoadjuvant setting, including in patients with sarcomatoid subtype. Aside from histology, other prognostic and predictive factors are currently under investigation, as PD-L1 has proved so far to be a less consistent biomarker for this disease. A plethora of studies investigating not only the role of other checkpoint molecules and pathways as well as the impact of adopted cellular therapies are also under way.

## Figures and Tables

**Table 1 ijms-25-10861-t001:** Select immunotherapy trials in advanced, previously treated pleural mesothelioma.

Study Name	Study Design	Number of Patients	Treatment	ORR (Study/Placebo)	PFS (Study/Placebo)	OS (Study/Placebo)
DETERMINE	Phase 2b Randomized	571 **	Tremelimumab vs. Placebo (2:1)	4.5/1.1% *	2.8/2.7 m	7.7/7.3 m
PROMISE-meso	Phase III Randomized	144	Pembrolizumab vs. chemotherapy	22/6% *	2.5/3.4 m	10.7/12.4 m
MESOT-TREM-2008	Phase 2 Single-Arm	29 **	Tremelimumab	6.9%	6.2 m	10.7 m
MESOT-TREM-2012	Phase 2 Single-Arm	29 **	Intensified Tremelimumab	3.4% 13.8% IR ORR	6.2 m IR PFS	11.3
KEYNOTE-028	Phase 1b Single-Arm	25	Pembrolizumab	20%	5.4 m	18 m
NCT02399371	Phase II, Single-Arm	65 **	Pembrolizumab	19%	4.5 m	11.5 m
KEYNOTE-158	Phase 2 Single-Arm	118	Pembrolizumab	8%	2.1 m	10 m
NivoMes	Phase II Single-Arm	34	Nivolumab	24%	2.6	11.8 m
MERIT	Phase II Single-Arm	34	Nivolumab	29%	6.1 m	17.3 m
CONFIRM	Phase 3 Randomized	332 **	Nivolumab vs. placebo	11/19% *	3/1.8 m *	10.2/6.9 m *
JAVELIN	Phase 1b	53 **	Avelumab	9%	4.1 m	10.7 m
MAPS2	Phase 2 Randomized	125	Nivolumab vs. Nivolumab + Ipilimumab	19/28%	4/5.6 m	11.9/15.9 m
INITIATE	Phase 2 Single-Arm	38	Ipilimumab + nivolumab	29%	6.2 m	NR

ORR = objective response rate; PFS = progression-free survival; OS = overall survival; * = *p* < 0.05; ** = study included both pleural and peritoneal mesothelioma; m = months; IR = immune-related; NR = not-reached.

**Table 2 ijms-25-10861-t002:** Select immunotherapy trials in advanced, treatment-naïve pleural mesothelioma.

Study Name	Study Design	Number of Patients	Treatment	ORR (Study/Placebo)	PFS (Study/Placebo)	OS (Study/Placebo)
NIBIT-Meso-1 *	Phase 2 Single-Arm	40 **	Tremelimumab + Durvalumab	28%	5.7 m	16.6 m
CheckMate743	Phase 3 Randomized	713	Nivolumab + Ipilimumab vs. Chemotherapy	40/43%	6.8/7.2 m	18.1/14.1 m *
DREAM	Phase 2 Single-Arm	55	Durvalumab + Chemotherapy	48%	6.9 m	18.4 m
PrE0505	Phase 2 Single-Arm	55	Durvalumab + Chemotherapy	56.4%	6.7 m	20.4 m
IND227	Phase III Randomized	440	Pembrolizumab + Chemotherapy vs. Chemotherapy	67%/47% *	7.13/7.16 m *	17.3/16.1 m *
BEAT-meso	Phase III Randomized	400	Atezolizumab + Chemotherapy + Bevacizumab vs. Chemotherapy + Bevacizumab	55/49%	9.2/7.6 m *	20.5/18.1 m

ORR = objective response rate; PFS = progression-free survival; OS = overall survival; * = *p* < 0.05; ** = study included pleural and peritoneal mesothelioma; m = months.

**Table 3 ijms-25-10861-t003:** Current neoadjuvant immunotherapy trials in resectable pleural mesothelioma.

Study Name	Identifier and Status	Phase	Number of Subjects	Histology	Neoadjuvant Intervention	Primary Objectives
Neoadjuvant Immune Checkpoint Blockade in Resectable Malignant Pleural Mesothelioma	NCT03918252 (Active, not recruiting)	I/II	30	Biphasic/ epithelioid	ARM A: Nivo Q2W× 3 * ARM B: Nivo Q2W× 3 + Ipilimumab × 1 dose *	Feasibility Safety
Testing the Addition of Immunotherapy Before Surgery for Patients with Sarcomatoid Mesothelioma	NCT05647265 (Recruiting)	II	26	Sarcomatoid or sarcomatoid- predominant biphasic	Nivolumab + Ipilimumab	Surgical rates PFS at 12 months
Induction Chemo+ Immunotherapy in Resectable Epithelioid and Biphasic Pleural Mesothelioma (CHIMERA Study)	NCT06155279 (Not yet recruiting) Italy only	II	40	All histologies	Pemetrexed-platinum-based chemotherapy + pembrolizumab every 3 weeks × 2 cycles	pCR
Neoadjuvant Durvalumab and Tremelimumab With and Without Chemotherapy for Mesothelioma	NCT05932199 (Recruiting)	I	52	All histologies	Cohort A: Durvalumab + tremelimumab × 3 cycles Cohort B: Platinum-pemetrexed + durvalumab + tremelimumab × 3 cycles	RFS

Nivo = nivolumab; Q2W = every 2 weeks; PFS = progression-free survival; pCR = pathologic complete response; RFS = recurrence-free survival; * Both arms to receive 1 year adjuvant nivolumab.

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
