# Peer review of "Immunotherapy for Treatment of Pleural Mesothelioma: Current and Emerging Therapeutic Strategies"

_ijms, 2024, doi:10.3390/ijms251910861_

Round 1
Reviewer 1 Report
Comments and Suggestions for Authors
The article provides a well-written overview of clinical trials (completed and ongoing) of immunotherapeutic approaches in pleural mesothelioma.
I have only few, mostly minor, comments:
While some sections end with a summarizing conclusion from the authors, this is missing in other sections, most notably after the section describing immunotherapy chemotherapy combinations. It would be good to learn more about the authors´ interpretation of the status of research from a concluding statement at the end of each main section of the manuscript.
Line 66: In many centers, the biphasic subtype seems to be more common than sarcomatoid PM, so that the statement “sarcomatoid mesothelioma is the second most common subtype” needs context.
Lines 66-78: The description of CDKN2A and MTAP misses the information that they are typically co-deleted and MTAP loss of expression is considered a surrogate marker for CDKN2A deletion.
Line 323: Bevacizumab targets VEGF not VEGF-R.
Sentences in lines 92-95 and 375-376 are unclear.
Author Response
Comments 1: While some sections end with a summarizing conclusion from the authors, this is missing in other sections, most notably after the section describing immunotherapy chemotherapy combinations. It would be good to learn more about the authors´ interpretation of the status of research from a concluding statement at the end of each main section of the manuscript.
Response 1: Thank you for suggesting this addition. We have added the following concluding statements:
- Line 271: The abundance of studies evaluating the role of immunotherapy for patients with recurrent or progressive mesothelioma have clearly shown a signal for response for a subset of patients, with some responders showing durable long-term responses. Given these findings, research has subsequently focused on attempting to utilize immunotherapy earlier in the disease process, identifying predictors of response to immunotherapy and novel combinations including strategies to enhance in the immune response. Table 1 summarizes key studies of immunotherapy for patients with progressive disease after initial treatment.
- Line 359: There is clearly a benefit with the use of immunotherapy in the first-line setting, with consistent improvements in survival seen compared with standard chemotherapeutic approaches. However, as with initial studies evaluating use for patients with refractory disease, further research into biomarkers of response to identify which patients benefit most, and strategies to enhance the immune response to treatment are needed. In addition, optimal combination strategies of immunotherapy with chemotherapy remain unclear. Table 2 summarizes key studies of immunotherapy in the first-line setting.
Comments 2: Line 66: In many centers, the biphasic subtype seems to be more common than sarcomatoid PM, so that the statement “sarcomatoid mesothelioma is the second most common subtype” needs context.
Response 2: Thank you for identifying this need for clarification. The sentence has been changed to “Sarcomatoid mesothelioma (SM) is diagnosed less often than epithelioid disease, and can be challenging to diagnosis as BAP1 loss is seen less frequently than in epithelioid disease” (line 66). In addition, line 72 was edited as follows: “Biphasic mesotheliomas are defined by disease involving at least 10% of both epithelioid and sarcomatoid components, with a true incidence rate being difficult to ascertain given the aforementioned diagnostic challenges.”
Comments 3: Lines 66-78: The description of CDKN2A and MTAP misses the information that they are typically co-deleted and MTAP loss of expression is considered a surrogate marker for CDKN2A deletion.
Response 3: Thank you for identifying this area for clarification. We have reworded line 68 to state: “In more than 90% of cases, sarcomatoid mesothelioma is found to have a CDKN2A homozygous deletion or loss of methylthioadenosine phosphorylase gene (MTAP), and in most cases co-deletion is seen. This is often identified by loss of expression of the MTAP gene protein product on immunohistochemistry as a surrogate marker for CDKN2A deletion, which can aid in making the diagnosis of SM.” We also added an additional reference (number 15).
Comments 4: Line 323: Bevacizumab targets VEGF not VEGF-R.
Response 4: Thank you for catching this mistake. Line 341 has been edited to state “Given previous findings from the MAPS trial using the VEGF targeted agent, bevacizumab, the phase III BEAT-meso study was designed”.
Comments 5: Sentences in lines 92-95 and 375-376 are unclear.
Response 5: Thank you for alerting us to this lack of clarity. Lines 95-99 reflect our change in the wording as follows: “Subsequently, a separate study evaluated the role of maintenance pemetrexed compared with observation alone after completion of four to six cycles of platinum chemotherapy and pemetrexed. 49 patients without progressive disease after this initial therapy were enrolled.” Line 410 was reworded to “CAR-T therapy involves creation of genetically modified T-cells which include a chimeric antigen receptor (or CAR); this receptor activates the T-cells against a tumor-specific antigen leading to an anti-tumor immune response.”
Reviewer 2 Report
Comments and Suggestions for Authors
Review article “Immunotherapy for Treatment of Pleural Mesothelioma: Current and Emerging Therapeutic Strategies” is well written.
Authors have covered most aspects of current and emerging therapeutic strategies Immunotherapy for Treatment of Pleural Mesothelioma.
Review article will consolidate the current information in this area and giving an update on the upcoming advancements in this area.
Major Comments
· Authors has not talked much on combined therapies being used in Pleural Mesothelioma, It should be discussed in more detail.
· It will be helpful to the readers if authors can consolidate Immunotherapies into a Table.
Minor Comments
1. Line 69- Italicize MTAP
2. Line 70 Delete “gene” MTAP protein.
3. Line 101 delete (OS) to make it consistent through the manuscript.
4. Line 252 Add coma after “respectively.”
5. Line 516 Italicize MTAP
6. Line 540 delete repeat 2024.
7. Check format of citations and delete repeated words.
Author Response
Comments 1: Authors has not talked much on combined therapies being used in Pleural Mesothelioma, It should be discussed in more detail.
Response 1: Thank you for this suggestion. Section 4.5 highlights dual immune-checkpoint inhibitor strategies studied to-date. Section 5 highlights dual immune-checkpoint strategies in the first-line setting, as well as combination strategies including immunotherapy and chemotherapy, as well as with the addition of VEGF-targeted therapy. The paragraph starting at line 349 highlights a combination strategy using an antibody-drug conjugate with pembrolizumab. In section 6, we highlight strategies including a small molecular inhibitor targeting VISTA as well as PD-1/PD-L1 pathway. There, we also discuss the role of LAG-3 targeted agents in combination with PD-1 antibodies, and have added additional studies starting at line 386 as follows: Targeting LAG-3 is an active area of investigation in the treatment of mesothelioma; the bispecific molecule tebotelimab which targets PD-1 and LAG-3 was studied in a phase 1 trial of patients with both solid tumors and hematologic cancers, including mesothelioma was found to be safe and warrants further investigation[61]. A phase 2 study evaluated the combination of ieramilimab, a humanized anti-LAG-3 antibody with spartalizumab (an anti-PD-1 antibody) for patients with select advanced solid malignancies, including mesothelioma[64]. 57 patients with mesothelioma were enrolled, and of those naïve to treatment with anti-PD-L1 or anti-PD-1 therapy, the ORR was 17.1% (90% CI 8.3-29.7%), with a median PFS of 5.5 months (Table 1).” Lastly, we added the following paragraph starting at line 493: “The phase II NIPU study randomized 118 patients with disease progression after first-line therapy to receive both nivolumab and ipilimumab alone, or combined with UV1, a vaccine directed against telomerase[91]. The rationale for this trial is to utilize a vaccine targeted towards a tumor-related antigen to increase lymphocyte infiltration and ultimately, immune response. The primary endpoint of PFS was not met, with a median PFS of 4.2 months for those in the study arm, compared with 4.7 months in the control arm (HR 1.01, 80% CI 0.75-1.36, p=0.979). However, there was a trend towards improved ORR (31% compared to 16%, OR 2.44, 80% CI 1.35-4.49, p=0.056) and overall survival (median 15.4 versus 11.1 months, HR 0.73, 80% CI 0.53-1.0, p=0.197).”
Comments 2: It will be helpful to the readers if authors can consolidate Immunotherapies into a Table.
Response 2: Thank you for this suggestion. As above, to that end, we have added 2 tables (now labeled table 1 and table 2, lines 280 and 366), which summarize the clinical trials involving immunotherapy which were discussed.
Comments 3: Line 69- Italicize MTAP
Comments 4: Line 70 Delete “gene” MTAP protein.
Comments 5: Line 101 delete (OS) to make it consistent through the manuscript.
Comments 6: Line 252 Add coma after “respectively.”
Comments 7: Line 516 Italicize MTAP
Comments 8: Line 540 delete repeat 2024.
Response 3-8: Thank you for identifying these grammatical errors. They have all been updated as requested.
Comments 9: Check format of citations and delete repeated words.
Response 9: Thank you, we have deleted repeated words and updated the references.
Reviewer 3 Report
Comments and Suggestions for Authors
The authors have provided a detailed summary of the treatment for malignant pleural mesothelioma. The cited references are appropriate, and I did not identify any major issues. I have one suggestion. I recommend creating a table for the study designs that include those related to neoadjuvant therapy and other treatments, including immune checkpoint inhibitors. I would appreciate your consideration.
Author Response
Comments 1: I recommend creating a table for the study designs that include those related to neoadjuvant therapy and other treatments, including immune checkpoint inhibitors. I would appreciate your consideration.
Response 1: Thank you for this suggestion. To that end, we have added 2 tables (now labeled table 1 and table 2, lines 280 and 366), which summarize the clinical trials involving immunotherapy which were discussed.